# Mouse Models of Peritoneal Carcinomatosis to Develop Clinical Applications

**DOI:** 10.3390/cancers13050963

**Published:** 2021-02-25

**Authors:** Ángela Bella, Claudia Augusta Di Trani, Myriam Fernández-Sendin, Leire Arrizabalaga, Assunta Cirella, Álvaro Teijeira, José Medina-Echeverz, Ignacio Melero, Pedro Berraondo, Fernando Aranda

**Affiliations:** 1Program of Immunology and Immunotherapy, Cima Universidad de Navarra, 31008 Pamplona, Spain; abellacarre@unav.es (Á.B.); cditrani@alumni.unav.es (C.A.D.T.); mfernandez.89@alumni.unav.es (M.F.-S.); larrizabala@alumni.unav.es (L.A.); acirella@unav.es (A.C.); ateijeiras@unav.es (Á.T.); imelero@unav.es (I.M.); 2Navarra Institute for Health Research (IDISNA), 31008 Pamplona, Spain; 3Bavarian Nordic GmbH, Fraunhoferstrasse 13, 82152 Planegg, Germany; jome@bavarian-nordic.com; 4Centro de Investigación Biomédica en Red de Cáncer (CIBERONC), 28029 Madrid, Spain; 5Department of Oncology, Clínica Universidad de Navarra, 31008 Pamplona, Spain; 6Department of Immunology and Immunotherapy, Clínica Universidad de Navarra, 31008 Pamplona, Spain

**Keywords:** peritoneal carcinomatosis, animal model, translational research, peritoneal microenvironment, metastasis

## Abstract

**Simple Summary:**

Peritoneal carcinomatosis mouse models as a platform to test, improve and/or predict the appropriate therapeutic interventions in patients are crucial to providing medical advances. Here, we overview reported mouse models to explore peritoneal carcinomatosis in translational biomedical research.

**Abstract:**

Peritoneal carcinomatosis of primary tumors originating in gastrointestinal (e.g., colorectal cancer, gastric cancer) or gynecologic (e.g., ovarian cancer) malignancies is a widespread type of tumor dissemination in the peritoneal cavity for which few therapeutic options are available. Therefore, reliable preclinical models are crucial for research and development of efficacious treatments for this condition. To date, a number of animal models have attempted to reproduce as accurately as possible the complexity of the tumor microenvironment of human peritoneal carcinomatosis. These include: Syngeneic tumor cell lines, human xenografts, patient-derived xenografts, genetically induced tumors, and 3D scaffold biomimetics. Each experimental model has its own strengths and limitations, all of which can influence the subsequent translational results concerning anticancer and immunomodulatory drugs under exploration. This review highlights the current status of peritoneal carcinomatosis mouse models for preclinical development of anticancer drugs or immunotherapeutic agents.

## 1. Introduction

Peritoneal Carcinomatosis (PCa) refers to the metastatic involvement of the peritoneum, characteristic of advanced-stage cancer [1]. PCa in gastrointestinal (e.g., colorectal cancer, gastric cancer) and gynecological (e.g., ovarian cancer) malignancies currently has a poor prognosis with a median survival of under six months [2]. At present, cytoreductive surgery in combination with hyperthermic intraperitoneal chemotherapy (HIPEC) has become widely accepted as an effective option to treat peritoneal metastasis [3,4]. However, locoregional HIPEC or pressurized intraperitoneal aerosol chemotherapy (PIPAC) treatments are only feasible in a small number of patients, and severe side-effects have been reported after peritoneal chemotherapy for PCa [5,6,7]. New therapeutic opportunities and experimental strategies in PCa are needed, and preclinical studies may help provide essential information about future successful clinical treatments.

Approaches based on mouse models allow researchers to learn about the tissue microenvironment in highly complex and dynamic physiological and pathological systems, such as cancer [8,9]. Clinical advances in cancer research as made over recent decades are linked to the efficient use of preclinical models of tumorigenesis. These constitute a critical tool to understand tumor growth and the interactions among different stromal and malignant cells in the tumor microenvironment. Indeed, mouse models of cancer are key to evaluating new therapeutic alternatives. In general, heterotopic syngeneic mouse models are subcutaneous grafts of same-strain tumor cells in mice due to the easy follow up of tumor growth. They represent a simple and accessible model, but suffer from the main disadvantage of not adequately reproducing the natural tumor microenvironment. This limitation is overcome by orthotopic engraftment (seeding tumor cells into the corresponding tissue), which offers a more realistic approach and provides a more relevant tissue microenvironment for assessing tumor development and treatment efficacy [10]. While syngeneic orthotopic models mimic more efficiently the disease biology of human disease, their main drawback relates to the monitoring of tumor progression, which requires the use of reporter genes, invasive surgical procedures or imaging methods such as real-time in-life fluorescence and bioluminescence imaging. Despite these limitations, syngeneic mouse models offer clues and proofs of concept in immunocompetent hosts and thereby represent an excellent preclinical platform to test compounds based on immuno-oncology targets to treat cancer [11].

Human tumor xenografted mouse models are also often used in preclinical cancer studies. Xenograft models constitute the cornerstone of the study of antineoplastic targeted compounds and tyrosine kinase inhibitors [12]. However, the lack of a fully functional immune system in these models is an obstacle in the study of the efficacy of most immunotherapy agents [13]. Furthermore, these models do not take into account cancer immunoediting, thus ignoring the role of the immune tumor microenvironment for cancer progression. Similar to traditional cell line-derived xenograft models, a more personalized option is the direct transfer of tumor specimens from individual patients, so-called patient-derived xenografts (PDXs) [14]. This model could also be engrafted with matched patient-derived peripheral blood mononuclear cells (PBMCs), making it a suitable translational research model for evaluating the efficacy of immunotherapeutic agents. Thus, results originating from PDX models are considered relevant for decision-making in drug development. However, the main drawbacks of the so-called humanized mice models are the limited source of tumor-initiating material, the long latency period for tumor development, and the high cost [13]. Moreover, some features such as the myeloid and vascular stromal compartments are not well reproduced. Furthermore, in addition to all the models on tumor cell inoculation into syngeneic, immunocompetent mice (suspension, tumor fragments, seeded 3D scaffolds, etc.), genetically engineered mouse models (GEMMs) also constitute useful experimental tools in cancer research [15,16]. GEMMs develop de novo tumors in a natural cancer immunoediting context. Oncogenes and tumor suppressor genes are modified in the somatic cells of mice leading to a classification based on the genetic intervention: Knock-out, knock-in, or transgenic mouse models [17,18,19,20]. Tumor onset in these models is sometimes accelerated by the administration of carcinogens or gene transfer of recombinases to activate the transforming tumorogenic and surrogate tumor antigens [21].

Despite all these models, establishing standardized, relevant, feasible, and reproducible PCa mice models remains a challenge. In general, peritoneal tumor burden or cancer progression in PCa mouse models can be quantified by (i) bioluminescence or biofluorescence: With the use of a tumor cell line engineered to express a reporter gene [22,23]; (ii) number and weight of peritoneal implants and/or omentum; iii) quantification of tumor burden by RT-PCR, immunohistochemistry, in vivo fluorescence microscopy, and flow cytometry, among other techniques [10,24]; (iv) ascites: Volume collected, hemorrhagic status; (v) survival: Mouse life span is evaluated by surveillance of spontaneous death or by euthanasia of animals with signs of pain or suffering according to established ethical protocol criteria [25,26]. Here, we provide a brief overview of reported mouse models to explore PCa in translational biomedical research (Figure 1).

## 2. PCa Mouse Models

### 2.1. PCa Syngeneic Models

A wide repertoire of syngeneic rodent cell lines has been studied in different PCa model research contexts. Among the most common cell lines to develop peritoneal metastasis in syngeneic mouse models are MC38 and CT26 (colon adenocarcinoma cell lines from the inbred C57BL/6 and Balb/c strains, respectively) [27,28,29,30,31,32], ID8 (an epithelial ovarian cancer cell line from C57BL/6) [33,34,35,36], YTN16 (a gastric cancer cell line from C57BL/6) [37,38], and Panc02 (a pancreatic adenocarcinoma epithelial cell line from C57BL/6) [25,39,40,41]. Some studies have also used the melanoma cell line B16.F10 for peritoneal carcinomatosis as an alternative albeit unreal model, in part as it offers the advantages of the implants being highly detectable due to melanin secretion and the aggressive progression of such tumors [42,43,44].

The classic PCa model consists of the inoculation of tumor cells directly into the peritoneal cavity. Depending on the tumor cell line and the degree of aggressiveness desired for the model (i.e., MC38: 2–5 × 10^5^ cells, and ID8: 5–10 × 10^6^ cells), the number of cells is critical, as well as a broad spread of volumes where cells are suspended for peritoneal inoculation (range from 400 to 700 µL) to disseminate cells throughout the cavity [27,28,29,33]. The development of observable peritoneal implants takes time. For ID8 PCa, this occurs around day 30–40 after injection of 5 × 10^6^ cells [10]. Several strategies have been used to increase the aggressiveness of the respective cell lines. Overexpression of vascular endothelial growth factor (VEGF) mediates increased tumor vascularization and correlates with poor prognosis. The ID8 cell line engineered to overexpress VEGF and, in some cases, the dendritic cells chemoattractant beta-defensin-29 (Defb29) has yielded dramatic increases in peritoneal implants and has reduced survival of the animals [35,45,46,47]. In addition, the overexpression of immunosuppressive molecules such as PD-L1 can promote PCa progression by inhibiting peritoneal cytotoxic lymphocytes [48].

Nevertheless, these kinds of peritoneal metastasis models do not reflect the biology underlying the onset of PCa since intraperitoneal (i.p.) inoculation spreads over the entire peritoneum. To address this, different studies have been carried out with a more realistic but laborious and often irreproducible model. This consists of generating orthotopic primary tumors (in the colon, ovary bursa, pancreas, etc.) and selecting cancer cells that have migrated to the secondary peritoneal organs localized as tumor implants or to the omentum after two or three weeks of primary tumor challenge. Then, consecutive passages are made in mice until a highly metastatic line is achieved. Using this selected aggressive cell line, PCa latency for experimental studies can be shortened [33,49,50,51,52]. The selection of these clones could also be useful for models such as ID8, where the development of peritoneal implants is very slow, even if 5 × 10^6^ cells are inoculated into the peritoneal cavity. By selecting the proper clone, this time can be significantly reduced to 20–30 days to obtain peritoneal implants after i.p. challenge.

Altogether, syngeneic PCa models are feasible and convenient models to test immunotherapeutic agents delivered into the peritoneal cavity. A major drawback is the significant differences between the peritoneum of rodents and humans. While the omentum, a highly vascularized organ critical in the development of PCa is a large organ in humans, it does not play the same role in mice [53] (Figure 2).

Despite this fact, the omentum of mice is considered an interesting organ for immune studies of peritoneal carcinomatosis, especially to understand the role of resident macrophages that promote peritoneal carcinomatosis [54,55,56]. However, on the whole, syngeneic PCa models are not entirely a bona fide model for PCa in patients.

### 2.2. PCa Human Xenograft and PDX Models

PCa xenograft models are based on the transplantation of human cancer cells or tissue into the peritoneal cavity of immunodeficient mice, such as athymic nude, non-obese diabetic (NOD), severe combined immunodeficiency (SCID), or NOD SCID gamma mice (NSG). These mice are unable to generate an immune response against human cells and tumor engraftment is thus promoted in the peritoneal cavity. The xenografted PCa models have been studied using colon cancer (HCT-116, LS174T, HT29, SW480, SW620, LoVo, RKO, Caco-2, KM12, MDST8, HUTU80) [57,58,59,60,61], pancreatic cancer (Panc-1, TCC-Pan2, BxPC3, AsPC-1) [62,63,64], gastric cancer (60As6, HSC-44PE, HSC-58, MKN45-P) [56,57,58,59,60,61,62,63,64,65,66,67,68,69], and ovarian cancer (OVCAR3, OVCAR4, OVCAR5, OVCAR8, CAOV3, OVSAHO, OV2944-HM-1, SKOV-3) [59,70,71,72] cell lines. These models have been used to attain proofs of concept and explore treatments that determine in vivo tumor cell cytotoxicity of drugs, such as chemotherapeutic agents. However, the main drawback of these standard PCa models is the lack of an immunocompetent environment [15]. In fact, xenograft models have been controversial due to their poor accuracy in predicting clinical response. For instance, they do not consider the effect of anti-tumor immune responses or the tumor immune microenvironment (tumor-infiltrating lymphocytes, macrophages, myeloid-derived immuno-suppressor cells, and regulatory T cells, among others) in the progression of cancer. Therefore, the use of PCa xenograft models is not useless to test immunotherapeutic agents but may lack an important component if adaptive immunity is involved in the mechanism of action.

To overcome the limitations of PCa xenograft models, PCa PDX models are currently under development. PDX models are ideally suited for testing potential, promising, and “personalized” cancer therapeutics [73]. However, although PDX models have been well-studied in different primary tumors (e.g., melanoma, lung cancer, breast cancer), peritoneal carcinomatosis remains to be an unexploited personalized platform to test cancer treatments. In line with this, Elien De Thaye and colleagues recently established and characterized PDX from peritoneal metastasis of ovarian cancer for the first time [74]. They used fresh peritoneal tissue specimens from 10 patients with metastatic ovarian cancer. These tumor fragments were processed to tumor slurry and inoculated by orthotopic engraftment i.p. into SCID/Beige mice. Then, tumors were harvested from the peritoneal implant established in the mouse by passages until they became more prominent. Overall, De Thaye et al. demonstrated a feasible orthotopic PDX model from a peritoneal metastasis of ovarian cancer and a sophisticated translational research platform. Furthermore, the generation of NSG mice has facilitated the co-engraftment of human hematopoietic cells, peripheral blood mononuclear cells (PBMCs), or bone marrow precursors [75]. The standard method to repopulate a mouse with human immune cells is to intravenously engraft PBMCs (autologous or allogenic) in immunodeficient mice. However, in the case of a PCa model, it would be more appropriate to isolate immune cells from peritoneum lavage or ascites from PCa patients and co-engraft these into the peritoneum of PDX mice with already established tumors. With this model, the local effects and activation of peritoneal resident immune cells could be studied with immunotherapeutic agents [76].

### 2.3. PCa Genetically Induced Models

Genetically engineered mouse models (GEMM) (transgenic, knock-out, knock-in mice) have been used to study peritoneal metastasis. GEMM can faithfully recapitulate some human cancers genetically with very similar tumor microenvironment phenotype [77,78]. Some mice models expressing human tumor endogenous antigens such as carcinoembryonic antigen (CEA) as a transgene have shown better engraftment of mice tumor cells expressing this antigen in particular, as is the case of the MC38-CEA adenocarcinoma cell line [79]. However, more elaborate and realistic models which more faithfully represent clinical peritoneal metastasis are achieved through gene modification of primary aggressive peritoneal tumors (i.e., ovary, colon, stomach, pancreas) induced to study early peritoneal metastasis. In line with, some studies have used triple-mutant (TKO) mice: p53^LSL-R172H/+^ Dicer1^flox/flox^ Pten^flox/flox^ Amhr2^cre/+^ [80,81]. This mouse p53^R172H^ is equivalent to human p53^R175H^ and is one of the most common p53 mutations in ovarian high-grade serous carcinoma (HGSCs) [82]. In TKO mouse, the tumors begin to form in the fallopian tube 1–2 months after birth. All mice develop PCa, with severe hemorrhagic ascites, leading to death (six months after birth).

Another recent and interesting work published by SSu-Hsueh Tseng, et al. reported on the development of a PCa model where the histological morphology and immune microenvironment are the same as peritoneal metastasis HGSCs in humans [83]. First, they observed that in immunocompetent mice the combination of shRNA-p53 (p53 suppression) with AKT and c-Myc oncogene overexpression in the peritoneum resulted in aggressive PCa with the presence of macroscopic peritoneal implants in only 21 days. This model consists of integrating the combination of shRNA-p53, AKT, and c-Myc via a sleeping beauty transposase system by i.p. administration, followed by electroporation in the abdominal cavity. Interestingly, this suppression of p53 and overexpression of AKT and c-Myc are able to overcome immunosurveillance and induce peritoneal tumors in immunocompetent mice too. Similarly, Sonia Iyer and colleagues constructed cell lines combining loss of Trp53 and overexpression of Ccne1, Akt2, and Trp53^R172H^, and driven by KRAS^G12V^ (KPCA) or Brd4 (BPCA) or Smarca4 (SPCA) overexpression [84]. Thus, this model represents an excellent and reliable platform for preclinical and translational PCa research and, more interestingly, the testing of immunotherapeutic agents. Additionally, this GEMM model can be used to investigate the initiation and progression of PCa, to identify potential biomarkers, and to predict the origin of peritoneal cancer spreading.

### 2.4. 3D-Biomimetics Peritoneal Implants

Advances in biotechnology and tissue engineering are increasingly being used in cancer research, as in other medical fields. One of these is the use of scaffolds as tools to test drugs or tumor implants with a structure that resembles the natural tumor [85]. Several studies have used these scaffolds to make an ex vivo tumor recapitulating the complexity of the tumor microenvironment. This has been achieved by including cancer cells, immune cells, fibroblasts, growth factors, and stromal components [86,87,88]. In PCa, this methodology has been used for the development of high-complexity bona fide peritoneal implants, although it is debatable whether this should be defined as an in vivo model per se. Therefore, this preclinical approach mimics the complexity (truly reflect in 3D models) of tumors developed in the peritoneum that may lead to potential therapies to improve the current PCa treatment options.

Accordingly, some groups are working on these 3D biomimetic peritoneal tumors. Daniela Loessner and colleagues observed that kallikrein-related peptidase 4, 5, 6, and 7 (KLK4-7) overexpression in ovarian cancer cells spheroids with integrin activation produces PCa after nine weeks of the inoculation in the peritoneal cavity in mice [88]. Moreover, in 2019, the same group recreated ex vivo ovarian tumor peritoneal metastasis. They used “3D co-cultured” hydrogel-encapsulated ovarian cancer cells with mesothelial cell-layered melt electrospun written scaffolds. These ex vivo tumor implants engrafted well and mimicked the same progression, invasion, and microenvironment as peritoneal implants of natural origin in mice [87].

Peritoneal implants present a complex structure, and the challenge of 3D culture is to mimic these ex vivo as realistically as possible so as to improve engraftment and the clinical biology in animal models. In order to achieve this, Emiel De Jaeghere et al. took into account the heterocellularity of 3D scaffolds in order to generate scaffolds which could then be implanted in the peritoneal cavity more successfully [86]. For that purpose, this group used polylactic acid (PLA) scaffolds with collagen type I hydrogel, co-seeding SKOV-3 cells (ovarian cancer), and cancer-associated fibroblasts (CAFs) not only to enhance the paracrine factor to improve spheroid formation in vitro, but also to enhance cancer cell survival, and angiogenesis. This model also seeks to improve the cellular composition and architecture of the tumor microenvironment and the organization of host cells in vivo. In fact, it was observed that CAFs, endothelial cells, macrophages, and cancer cells showed similar features when compared with patient-derived peritoneal metastasis. The authors concluded that this scaffold model represents a promising platform for the preclinical study of drug penetration and efficacy following delivery of intraperitoneal chemotherapy, among other therapeutic agents [86].

## 3. Concluding Remarks

Overall, PCa models have both advantages and drawbacks (Figure 3). Therefore, the read-out of the results obtained in mice should always be interpreted with caution. The complexity of cancer development and all the biological factors involved make it even more difficult to construct an ideal preclinical syngeneic model. Even PDX models have sometimes been shown not to correlate with the therapeutic outcome observed in the same patient from whom the model was derived.

In general terms, the human peritoneum environment where PCa develops is highly dynamic and is, at least currently, unmatched in mice with the same tissue structure and biology. However, syngeneic, humanized, personalized PDX, GEMM models, or approaches using biotechnology for 3D tumors have offered proofs of concept enabling the preclinical study of promising therapeutic alternatives in patients with PCa. Unfortunately, the currently dismal survival figures for PCa will only become worse in the following years due to the collateral COVID-19 impact associated with the cancellation of multiple routine medical exams to detect early gastrointestinal or gynecologic primary tumors. For this reason, exploring new PCa animal models as a platform to test, improve and/or predict the appropriate therapeutic interventions in patients should be a cancer research priority in the coming years.

## Figures and Tables

**Figure 1 cancers-13-00963-f001:**
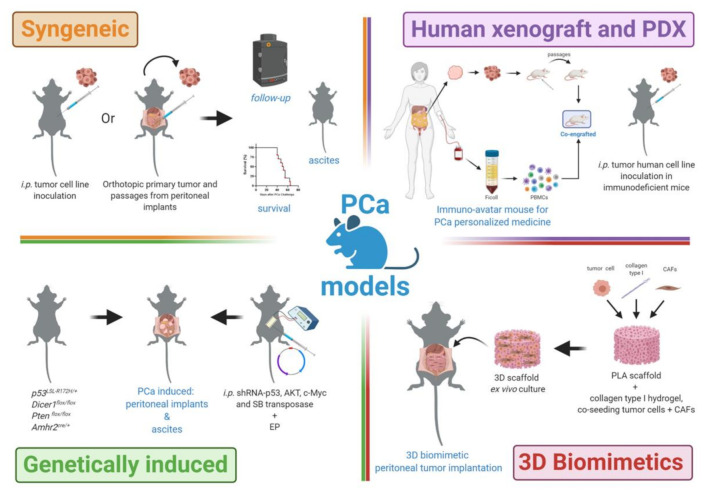
Peritoneal carcinomatosis mouse models. A repertoire of preclinical platforms to test new therapeutic opportunities and transfer the results to patients with peritoneal carcinomatosis. CAFs, cancer-associated fibroblasts; EP, electroporation; i.p., intraperitoneally; PBMCs, peripheral blood mononuclear cells; PCa, peritoneal carcinomatosis; PDX, patient-derived xenograft; PLA, polylactic acid; SB, sleeping beauty. Created with BioRender.com, accessed on 19 February 2021.

**Figure 2 cancers-13-00963-f002:**
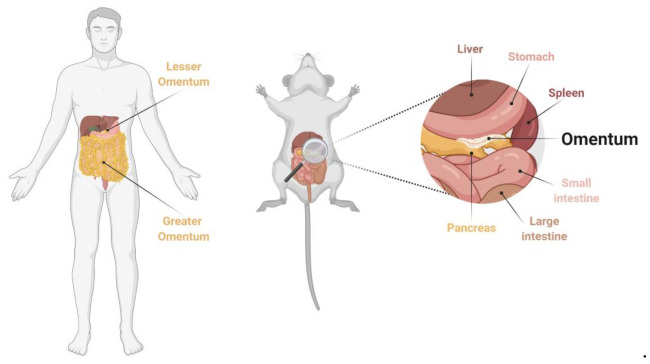
Anatomy and localization of human and mouse omentum. The mouse omentum is a thin, slightly elongated, and vascularized tissue located under the stomach and between the spleen and pancreas. Created with BioRender.com, accessed on 19 February 2021.

**Figure 3 cancers-13-00963-f003:**
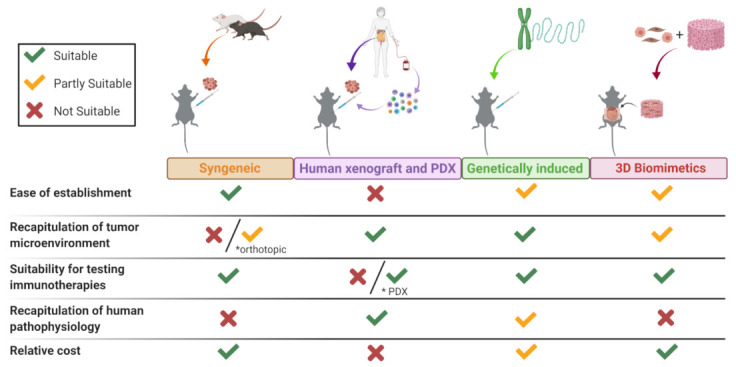
Benefits and limitations of PCa mouse models. Assessment of the current platforms for preclinical development of anticancer drugs or immunotherapeutic agents in PCa. Relative scores are represented as being suitable (green checkmark), partially suitable (yellow checkmark), and not suitable (red cross). Created with BioRender.com, accessed on 19 February 2021.

## Data Availability

Not applicable.

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
