# Peer review of "Mouse Models of Peritoneal Carcinomatosis to Develop Clinical Applications"

_cancers, 2021, doi:10.3390/cancers13050963_

Round 1

Reviewer 1 Report

The review manuscript highlights the current status of peritoneal carcinomatosis mouse models for preclinical development of anticancer drugs and immunotherapeutic agents. It provides a concise overview of the different mouse models, it is well-written, nicely structured and clear. The visual information is very helpful and useful. It lists the pros and cons of the different models especially with a view on translating results to the human situation.

Major comments

General: the whole article is focused on mouse models, and I would like to understand why? There are in the mean time also good rat models available, that can be used to study certain treatment options, but nothing is said about them. What was the rationale behind the authors’ choice?

Line 32: Is PIPAC indeed already widely accepted as a treatment option? This seems to be a bit of an overstatement from my perspective. Please rephrase.

Section 2.1 and 2.2: Please include a discussion of the cell types used in these models, and how representative they are for the typical observed human tumor metastases in the peritoneal cavity. Can one obtain reliable information from such aggressive tumor models that are predictive for therapeutic effect sizes?

Minor comments

Line 13 - Abstract: Indicate the primary tumors that can have metastases in the peritoneal cavity. I think that that is relevant info to add here.

Line 28: Are gastric cancers also not possible? Better replace ‘i.e.’ by ‘e.g.’ for both examples

Line 34: ‘.. and severe side-effects have been reported after peritoneal chemotherapy for PCa.’ Please add a reference to this statement.

Line 47: ‘orthotopic-engraftment’: explain what this is so that the reader is clear and doesn’t have to look elsewhere for an explanation

Line 48: provide -> provides

Line 51: monitorization -> monitoring

Line 148: syngenic -> syngeneic

Line 159: , and -> drop the comma

Line 214: overecome -> overcome

Line 222: 2.4.3. D biomimetics peritoneal implants -> 3D biomimetics peritoneal implants. It strikes me that this section is less well written than the other ones. Please carefully review once more.

Lines 231-232: Therefore, pre- or clinically predictive approach mimics the complexity (truly reflect in 3D models) of tumors developed in the peritoneum that may lead to potential therapies to improve the current PCa treatment options. This sentence is not very clear to me. Please rephrase – what do you intend to say here?

Line 246: 3D scaffold -> 3D scaffolds

Line 256: study drug of penetration -> study of drug penetration

Author Response

Reviewer 1

We would like to thank you for your careful and thorough reading of this manuscript and for all the comments and constructive suggestions, which undoubtedly help to improve the quality of this manuscript. Please find below a point-by-point response to each of the reviewer´s commentaries.

Major comments

General: the whole article is focused on mouse models, and I would like to understand why? There are in the mean time also good rat models available, that can be used to study certain treatment options, but nothing is said about them. What was the rationale behind the authors’ choice?

We do agree with the reviewer´s annotation that good rat models (WAG/Rij and BD IX Rat strains) in peritoneal carcinomatosis are available (Intraoperative hyperthermic intraperitoneal chemotherapy after cytoreductive surgery for peritoneal carcinomatosis in an experimental model. Klaver YL, et al., 2010; A comparison between radioimmunotherapy and hyperthermic intraperitoneal chemotherapy for the treatment of peritoneal carcinomatosis of colonic origin in rats. Aarts F, et al., 2007). Even have been studies in large animal models such as sheep (Orthotopic xenografts of human melanoma and colonic and ovarian carcinoma in sheep to evaluate radioimmunotherapy. Turner JH, et al., 1998) or pigs (Intraperitoneal cell movement during abdominal carbon dioxide insufflation and laparoscopy – an in vivo model. Hewett PJ, 1996). Nonetheless, we have focused this manuscript on the mouse model without discriminating alternative peritoneal carcinomatosis animal models. We have chosen mice for expertise and knowledge matters.

Line 32: Is PIPAC indeed already widely accepted as a treatment option? This seems to be a bit of an overstatement from my perspective. Please rephrase.

We thank the reviewer´s commentary. We have decided to remove PIPAC treatment option in this context as a widely accepted peritoneal carcinomatosis treatment option. PIPAC is also applied in the treatment of peritoneal carcinomatosis, but in more advanced stages of cancer when debulking surgery cannot be performed. The revised version of the manuscript will read as follows: “At present, cytoreductive surgery in combination with hyperthermic intraperitoneal chemotherapy (HIPEC) has become widely accepted as an effective option to treat peritoneal metastasis”

Section 2.1 and 2.2: Please include a discussion of the cell types used in these models, and how representative they are for the typical observed human tumor metastases in the peritoneal cavity. Can one obtain reliable information from such aggressive tumor models that are predictive for therapeutic effect sizes?

We thank the reviewer´s commentary. Accordingly, we have mentioned that PCa in syngeneic and human xenograft or PDXs models can offer only proofs-of-concept enabling the pre-clinical study of promising therapeutic alternatives in patients with PCa. We have discussed this issue in concluding remarks.

Minor comments

Line 13 - Abstract: Indicate the primary tumors that can have metastases in the peritoneal cavity. I think that that is relevant info to add here.

We thank the reviewer suggestion in the Abstract section. Accordingly, we have include primary tumors origin as examples which will read as follows in the revised version of the manuscript: “Peritoneal carcinomatosis of primary tumors origin in gastrointestinal (e.g., colorectal cancer, stomach cancer) or gynecologic (e.g., ovarian cancer) malignancies is widespread tumor dissemination in the peritoneal cavity for which few therapeutic options are available.”

Line 28: Are gastric cancers also not possible? Better replace ‘i.e.’ by ‘e.g.’ for both examples

We thank the reviewer´s commentary. Accordingly, we have included gastric cancer in gastrointestinal malignancies and replaced i.e. by e.g.

Line 34: ‘.. and severe side-effects have been reported after peritoneal chemotherapy for PCa.’ Please add a reference to this statement.

We thank the reviewer´s indication. We have included 3 new references about complications and toxicities after peritoneal chemotherapy for PCa.

Line 47: ‘orthotopic-engraftment’: explain what this is so that the reader is clear and doesn’t have to look elsewhere for an explanation

We appreciated the reviewer´s commentary. We have specified and the final version of the manuscript will read as follows: “This limitation is overcome by orthotopic-engraftment (seeding tumor cells into the corresponding tissue)”

Line 48: provide -> provides

We thank the reviewer´s notification. The error has been corrected.

Line 51: monitorization -> monitoring

We thank the reviewer´s notification. The error has been corrected.

Line 148: syngenic -> syngeneic

We thank the reviewer´s notification. The typographic error has been corrected.

Line 159: , and -> drop the comma

We thank the reviewer´s notification.

Line 214: overecome -> overcome

We thank the reviewer´s notification. The typographic error has been corrected.

Line 222: 2.4.3. D biomimetics peritoneal implants -> 3D biomimetics peritoneal implants. It strikes me that this section is less well written than the other ones. Please carefully review once more.

We thank the reviewer´s recommendation. We have changed the typographic mistake and reviewed this section.

Lines 231-232: Therefore, pre- or clinically predictive approach mimics the complexity (truly reflect in 3D models) of tumors developed in the peritoneum that may lead to potential therapies to improve the current PCa treatment options. This sentence is not very clear to me. Please rephrase – what do you intend to say here?

Following the reviewer’s suggestion, the sentence has been modified and now will read as follows: “Therefore, this preclinical approach mimics the complexity (truly reflect in 3D models) of tumors developed in the peritoneum that may lead to potential therapies to improve the current PCa treatment options.”

Line 246: 3D scaffold -> 3D scaffolds

We thank the reviewer´s notification. The error has been corrected.

Line 256: study drug of penetration -> study of drug penetration

We thank the reviewer´s notification. The error has been corrected.

Reviewer 2 Report

In this review, Bella et al. summarise the different pre-clinical mouse models to study peritoneal carcinomatosis. This review is timely and deserves to be published in Cancers.

Just a couple of minor comment:

Line 44: Authors stated “syngeneic mouse models are subcutaneous grafts”. The term syngeneic model is applicable to other tumour sites, not only the subcutaneous site. Therefore, this sentence should be corrected.

Line 259: Figure instead of Figue

Author Response

Reviewer 2

In this review, Bella et al. summarise the different pre-clinical mouse models to study peritoneal carcinomatosis. This review is timely and deserves to be published in Cancers.

We thank the reviewer for the positive comments made on our manuscript. Following the reviewer’s indication, we have answered the two minor observations.

Just a couple of minor comment:

Line 44: Authors stated “syngeneic mouse models are subcutaneous grafts”. The term syngeneic model is applicable to other tumour sites, not only the subcutaneous site. Therefore, this sentence should be corrected.

We thank the reviewer´s commentary. We do agree with the reviewer, and we have clarified the sentence. The final version of the manuscript will read as follows: “In general, heterotopic syngeneic mouse models are subcutaneous grafts of same-strain tumor cells in mice due to the easy follow up of tumor growth”.

Line 259: Figure instead of Figue

We thank the reviewer´s observation. The typographic error has been corrected.

Reviewer 3 Report

Bella and colleagues are presenting an overview of  pre-clinical models assessing peritoneal carcinomatosis. They include a description of several models, relevant literature, and discuss strengths and weaknesses.

Overall, the manuscript is very well written and well introduced. It has a focused message and helpful and esthetic looking figures. I only have a few minor comments or suggestions:

  • For readers it would be helpful to have more references included, especially when it comes to various cell lines that are being used. Eg. in line 112, and lines 156-158.
  • In the table of figure 3: patophysiology should be pathophysiology
  • In line 175, please explain why PCa remains to be a challenge as a personalized platform to test cancer treatments. 
  •  Please explain what CEA (line 196) stands for.
  • Could the authors include the timeline how long it takes for PCa to take place in the models they describe? For exmaple, it's not completely clear if the TKO model gives rise to PCa, and in which timeframe.
  • The title of 2.4.3 (line 222) seems a bit odd
  • the sentence at line 231-233 is a bit strange; Are scaffolds meant with pre-or clinically predictive approach? I'm a it confused what you are trying to say here.
  •  

Author Response

Reviewer 3

Bella and colleagues are presenting an overview of  pre-clinical models assessing peritoneal carcinomatosis. They include a description of several models, relevant literature, and discuss strengths and weaknesses.

Overall, the manuscript is very well written and well introduced. It has a focused message and helpful and esthetic looking figures. I only have a few minor comments or suggestions:

We thank the reviewer for the positive comments made on our manuscript. Following the reviewer’s indication, we have answered below point-by-point.

For readers it would be helpful to have more references included, especially when it comes to various cell lines that are being used. Eg. in line 112, and lines 156-158.

We do agree with the reviewer´s comment, and we have included more references.

In the table of figure 3: patophysiology should be pathophysiology

We thank the reviewer´s observation in Figure 3. The typographic error has been corrected.

In line 175, please explain why PCa remains to be a challenge as a personalized platform to test cancer treatments.

After rereading the sentence, we have changed challenge by a term more appropriate:  “unexploited”. The final version of the manuscript will read as follows: “However, although PDX models have been well-studied in different primary tumors (e.g., melanoma, lung cancer, breast cancer), peritoneal carcinomatosis remains to be an unexploited personalized platform to test cancer treatments.”

Please explain what CEA (line 196) stands for.

We thank the reviewer´s observation. We have defined the abbreviature of CEA, carcinoembryonic antigen, in the final version of the manuscript.

Could the authors include the timeline how long it takes for PCa to take place in the models they describe? For exmaple, it's not completely clear if the TKO model gives rise to PCa, and in which timeframe.

We thank the reviewer´s suggestion. According to TKO model examinations, tumors begin to form in the fallopian tube between 1–2 months after birth. All mice develop PCa, with severe hemorrhagic ascites, leading to death (6 months after birth), but is not reported the timeline of how long it takes for the onset of PCa. We have included this information in the final version of the manuscript.

The title of 2.4.3 (line 222) seems a bit odd

We thank the reviewer´s observation in Figure 3. The typographic error has been corrected and will read as follows: “3D biomimetics peritoneal implants”

the sentence at line 231-233 is a bit strange; Are scaffolds meant with pre-or clinically predictive approach? I'm a it confused what you are trying to say here.

We have clarified the sentence and now will read as follows: “…scaffold model represents a promising platform for the preclinical study of drug penetration and efficacy following delivery of intraperitoneal chemotherapy, among other therapeutic agents.”

Reviewer 4 Report

This review has been well written and worth to publish in “Cancers”. However, there is one point to be modified.

  1. In chapter 2.2., line 158: YTN16 cells are derived from C57BL/6 mice gastric cancer cells, not from human gastric cancer. PC model inoculated YTN16 alone showed solid tumor without stroma. Fujimori et al. (Ref. No.58) demonstrated that PC allograft model co-inoculated with YTN16 and mouse myofibroblasts LmcMF cells after co-culture these cells showed fibrous tumor with activated CAFs, resulting high M2 macrophages infiltration and low CD8 positive cell infiltration, which resemble to clinical features in tumor (immune) microenvironments of human PC.

This model should be the most useful PC animal models as a gastric cancer. Please consider to describe above contents in your manuscript, not delete reference No. 58.

Author Response

This review has been well written and worth to publish in “Cancers”. However, there is one point to be modified.

We thank the reviewer for the positive comment made on our manuscript. Following the reviewer’s observation, we have answered below.

In chapter 2.2., line 158: YTN16 cells are derived from C57BL/6 mice gastric cancer cells, not from human gastric cancer. PC model inoculated YTN16 alone showed solid tumor without stroma. Fujimori et al. (Ref. No.58) demonstrated that PC allograft model co-inoculated with YTN16 and mouse myofibroblasts LmcMF cells after co-culture these cells showed fibrous tumor with activated CAFs, resulting high M2 macrophages infiltration and low CD8 positive cell infiltration, which resemble to clinical features in tumor (immune) microenvironments of human PC.

This model should be the most useful PC animal models as a gastric cancer. Please consider to describe above contents in your manuscript, not delete reference No. 58.

We very much appreciate the reviewer´s observation and explanation about YTN16 cell line and its origin. We have changed YTN16 PCa gastric model in the syngeneic model's section.
